# A Composite Fabric with Dual Functions for High-Performance Water Purification

**DOI:** 10.3390/ma15175917

**Published:** 2022-08-26

**Authors:** Yankuan Tian, Xin Yang, Long Xu, Xueli Wang, Jianyong Yu, Dequn Wu, Faxue Li, Tingting Gao

**Affiliations:** 1Key Laboratory of Textile Science & Technology, Ministry of Education, College of Textiles, Donghua University, Shanghai 201620, China; 2Innovation Center for Textile Science & Technology, Donghua University, Shanghai 201620, China

**Keywords:** water pollution, solar-vapor generation, photocatalytic, dyeing wastewater

## Abstract

The dilemma of diminishing freshwater resources caused by water pollution has always impacted human life. Solar-driven interfacial evaporation technology has the potential for freshwater production via solar-driven distillation. However, in solar-driven interfacial evaporation technology, it is difficult to overcome the problem of wastewater containing various contaminants. In this work, we propose a bifunctional fabric created by depositing titanium dioxide@carbon black nanoparticles onto cotton fabric (TiO_2_@CB/CF). The TiO_2_@CB/CF has a coupling effect that includes the photothermal effect of CB and photocatalysis of TiO_2_, and it can not only generate clean water but can also purify contaminated water. The resulting bifunctional fabric can achieve an outstanding water evaporation rate of 1.42 kg m^−2^ h^−1^ and a conversion efficiency of 90.4% in methylene blue (MB) solution under one-sun irradiation. Simultaneously, the TiO_2_@CB/CF demonstrates a high photocatalytic degradation of 57% for MB solution after 2 h with light irradiation. It still shows a good photocatalysis effect, even when reused in an MB solution for eight cycles. Furthermore, the TiO_2_@CB/CF delivers excellent performance for actual industrial textile dyeing wastewater. This bifunctional fabric has a good application prospect and will provide a novel way to resolve the issue of freshwater scarcity.

## 1. Introduction

Over-discharging contaminants into natural water resources causes serious environmental pollution and has led to a global shortage of fresh water. It is reported that, around the world, approximately 420 billion cubic meters of sewage are discharged into the environment annually. Thus, it is urgent to address the water pollution problem [1,2,3]. To date, many important technologies, such as reverse osmosis (RO) [4,5] and thermal desalination technologies [6,7], have been widely used to generate fresh water from wastewater. However, due to these traditional techniques having high expense, complex technical requirements, and fossil energy consumption, a novel, inexpensive, and low-energy-intensive technique should be developed. Solar distillation is an ancient and clean technology that utilizes inexhaustible solar energy to achieve the evaporation of water, and then collects the condensate to produce clean water [8,9,10]. Nevertheless, the shortcomings of low conversion efficiency have always hindered its application and development. To solve the dilemma of solar distillation, solar-driven interfacial evaporation technology, which localizes heat on the air–water interface, has been developed and has recently attracted more and more attention [11,12,13,14,15,16].

Recently, a series of photothermal materials, such as metal nanoparticles [17,18], carbon-based materials [19,20], and polymers [21,22], have been applied to solar-driven interfacial evaporation technology. Li et al. designed a three-dimensional artificial interfacial evaporator that could provide a simple way for wastewater treatment with carbon materials, achieving the recovery of heavy metals and the production of purified water [23]. Wang et al. used MXene, carbon nanotubes, and cotton fabric to fabricate an interfacial solar-vapor generation material for textile wastewater purification. It can obtain an evaporation rate of 1.16 kg m^−2^ h^−1^ for textile wastewater under one sun illumination and reduce the concentration of organic–inorganic pollutants in concentrated fresh water [24]. Nevertheless, there are still many problems when sewage is used as the water source for solar-driven interfacial evaporation: (1) although solar-driven interfacial evaporation technology can generate freshwater, it cannot solve the problem of pollutants that are unfavorable to the environment being contained in the raw water, such as organic dyes and bacteria; (2) due to the generation of steam, the concentration of raw water was increased, so the efficiency of solar-vapor generation gradually decreased. Therefore, developing a novel approach for continuous highly-efficient evaporation and decontamination of polluted water to produce fresh water is highly desirable.

Photocatalytic technology is an effective approach to degrade dyeing byproducts for collecting clean water. Some photoactive materials absorb solar energy to create highly active species. Thus, some small non-toxic molecules can be produced when the highly-active species react with contaminants [25,26,27,28]. Therefore, it is a creative strategy to integrate the photocatalytic nanomaterials into interfacial photothermal materials. Importantly, the synergy between solar-vapor generation and photocatalysis has been demonstrated: on the one hand, photocatalyst can effectively solve the problem of pollutants in the process of solar-vapor generation; on the other hand, through the coupling effect between photothermal materials and catalyst, the absorption band of the catalyst is increased, thus improving the catalytic effect [29,30,31,32]. Liu et al. prepared a multifunctional membrane, which was divided into three layers from top to bottom: a layer of TiO_2_ nanoparticles (NPs), a layer of Au NPs, and a layer of anodized aluminum oxide (AAO). With the irradiation of simulated solar energy, the functional membrane not only purified polluted water through photocatalytic degradation but also produced clean water through evaporation [31]. Hao et al. prepared a bifunctional cotton fabric with both the photothermal effect of pyrrole (PPy) and the photocatalytic properties of titanium dioxide (TiO_2_) nanoparticles. With simulated solar irradiation, the TiO_2_-PDA/PPy/cotton fabric showed an excellent photocatalysis effect and achieved a ~96% methyl orange (MO) degradation over 3 h [32]. Nevertheless, studies should be conducted among these works to address some limitations. Firstly, a system with less expensive materials and simpler fabrication processes should be developed. Secondly, when simulated dyeing water is used as the water source, the coupling effect of the solar-driven evaporation and photocatalysis should be explored in detail with solar light irradiation. Meanwhile, few studies about actual industrial wastewater have been carried out, resulting in an unclear practical application of the bifunctional materials.

As a widely-used photocatalyst, TiO_2_ nanoparticles are low-cost, stable, and environmentally friendly. However, due to the large bandgap (anatase, ~3.2 eV) of TiO_2_, high photocatalytic efficiency is difficult to achieve [33,34,35,36,37,38,39]. To solve the problem, material engineering is usually used to increase the absorption range and expand the usage of photons in the range outside that of the catalytic absorption band. The methods used include doping with metal or non-metals [31,32], compounding with semiconductors [33], sensitization with organic dyes [34], and co-doping with carbon materials [35]. In this work, we prepared a novel cotton fabric that couples solar-driven evaporation and photocatalysis, which expands the usage of solar energy and increases the efficiency of wastewater treatment. The bifunctional fabric is fabricated by depositing titanium dioxide@carbon black nanoparticles onto the cotton fabric (TiO_2_@CB/CF, Figure 1). The TiO_2_@CB/CF enables synergistic effects of solar-driven evaporation and photocatalysis: due to the fast water transport of hydrophilic cotton fabric [40,41,42] and the photothermal effect of CB, the TiO_2_@CB/CF can generate clean water through localized heating on the air–water interface; with sunlight irradiation, the photocatalytic function of the TiO_2_@CB/CF can degrade the contaminants of sewage, and it is worth noting that the photothermal effect of CB can accelerate the photocatalytic effect of TiO_2_. The resulting bifunctional fabric can achieve an outstanding water evaporation rate of 1.42 kg m^−2^ h^−1^ and a conversion efficiency of 90.4% in MB solution under one-sun irradiation. Additionally, a degradation of 57% for MB solution is obtained. Furthermore, when the actual industrial dyeing wastewater is used as raw water, the TiO_2_@CB/CF can generate freshwater and optimize the water quality. This bifunctional fabric expands the application range of interfacial solar-vapor generation materials and has good prospects in the practical application of sewage treatment.

## 2. Materials and Methods

### 2.1. Materials

Cotton fabric with plain weave was bought from Jialuolan Trading Co., Ltd., Jiangsu, China. Tetrabutyl titanate and ethyl alcohol were obtained from Macklin Industrial Co., Ltd., Shanghai, China. NaCl was purchased from Aladdin Industrial Co., Ltd., Shanghai, China. CB nanoparticles were supplied from Tianjin Tianyi Century Chemical Products Technology Development Co., Ltd., Tianjin, China. MB was obtained from SINOPHARM chemical reagents, Chongqing, China. Industrial dyeing wastewater was offered from Shenghong Dyeing and Finishing Co., Ltd., Suzhou, China.

### 2.2. Synthesis of TiO_2_ Nanoparticles

The sol-gel method was used to prepare the TiO_2_ nanoparticles. Deionized water (DI) of 50 mL was used as solution A, and the mixed solutions with ethanol of 5 mL and tetrabutyl titanate of 20 mL were used as solution B. Solution A was then added slowly to solution B while solution B was stirred continually. Some white solid substances then separated from solution B. The white solid substances were dried at 100 °C for 24 h and pulverized into solid powder. Finally, the TiO_2_ nanoparticles were obtained.

### 2.3. Synthesis of TiO_2_@CB Nanoparticles

A dispersion of CB (3 wt%) was used as solution A to replace the DI in Section 2.2, and the TiO_2_@CB nanoparticles were obtained by the same process as in Section 2.2.

### 2.4. Preparation of TiO_2_@CB/CF

The TiO_2_@CB nanoparticles were uniformly dispersed in dispersion liquid by a sonication machine. The TiO_2_@CB nanoparticles were then evenly distributed on the cotton fabric by filtrating. Finally, the TiO_2_@CB/CF was prepared. Similarly, the TiO_2_/CF and CB/CF can be prepared by the same method.

### 2.5. Characterization

Scanning electron microscopy (SEM, SU8010, HITACHI, Hitachi, Japan) was performed to observe the chemical element of the TiO_2_@CB/CF. Fourier-transform infrared (FTIR, NEXUS-670, GNI, Woburn, MA, USA) spectra were used at room temperature from 400 to 4000 cm^−1^. An ultraviolet-visible-near-infrared spectrometer (UV-vis-NIR, UV 3600, Shimadzu Scientific Instruments, Kyoto, Japan) was used to test the total transmittance (*T*) and reflectance (*R*) of a sample. The water contact angles of different materials were obtained with a goniometer (OCA40, Dataphysics, Filderstadt, Germany) using 2 μL of water at ambient temperature.

### 2.6. Evaporation under Simulated Solar Irradiation

A solar simulator (Beijing Zhongjiao Jinyuan Technology Co., Ltd., Beijing, China) was used to simulate a light source (~1 kW m^−2^). The MB solution (5 mg L^−1^) in the beaker was used as the original solution, and the sample floated above the liquid. An electronic balance (ME204, METTLER TOLEDO, Columbus, OH, USA) was used to monitor the weight change of the whole device in real-time, and an IR camera (T630sc, FLIR, Wilsonville, OR, USA) and thermocouple (HH506RA, Omega, New York, NY, USA) were used to record bulk temperatures. An ambient temperature of 25 °C and ambient humidity of 60% were recorded.

### 2.7. Photocatalytic Activity Test

To evaluate the photocatalytic activity, an organic MB solution (5 mg L^−1^) was used to test the photocatalytic properties of the sample. Similarly, the experiment device of the above evaporation experiment was suitable for catalysis experiments. The absorbance spectrums of the different samples were measured every 30 min using a UV−vis spectrometer. The concentration of MB solution was determined by the absorbance at 664 nm. The degradation ($D_{e}$) of the MB solution can be characterized by the following equation:$$D_{e}=\frac{C_{0}-C}{C_{0}}\times 100\%$$
where $C_{0}\;$and C are the initial concentration and the concentration at the irradiation time of *t* of the MB solution. For the authenticity of the experiment, every sample was tested at least three times.

## 3. Results and Discussions

The distribution of TiO_2_@CB nanoparticles on the cotton fabric is vital for the performance of TiO_2_@CB/CF. From Figure 2a,b, we can see that the TiO_2_@CB nanoparticles are firmly attached to the surface of the cotton fabric and evenly distributed. The bonding between nanoparticles and fabric is beneficial for the durability of the bifunctional cotton fabric. Figure 2c shows the morphology of the TiO_2_@CB nanoparticles. We can see that hundreds of CB nanoparticles with a diameter of ~1 nm surround TiO_2_ particle with a diameter of ~600 nm. The interaction between the TiO_2_ and CB nanoparticles is achieved through Van der Waals forces, and they form secondary structures called aggregates [35]. Similarly, from Appendix A, we can also observe the morphologies of TiO_2_/CF and CB/CF. To further demonstrate the distribution of TiO_2_@CB nanoparticles on cotton fabrics, the elements of the TiO_2_@CB/CF were observed by EDS mapping images (Figure 2d). We can see that various elements, including Ti, cover the cotton fabric and are evenly distributed, which is beneficial for the coupling effect of TiO_2_ and CB [32].

Figure 2e demonstrates the chemical characterization of the three cotton fabrics by FTIR spectra. The existence of absorption peaks at 1730 and 2236 cm^−1^ shows that the CB nanoparticles successfully attach to the surface of the cotton fabric, and the characteristic peaks at 597 and 1569 cm ^−1^ can be attributed to the existence of Ti-O, which can improve the hydrophilicity of the TiO_2_/CF and TiO_2_@CB/CF. Moreover, the broad absorption peak from 978 to 1365 cm^−1^ corresponds to the characteristic peak of Ti-O-C in TiO_2_@CB/CF, which indicates that the TiO_2_@CB nanoparticles have been successfully covered on the cotton fabric [43,44,45,46]. The hydrophilicity of interfacial materials has been proved to be a vital approach to improving the solar-vapor conversion efficiency [47,48]. Due to the existence of the Ti-O, the contact angle of TiO_2_@CB/CF is 39° (Figure 2f), which shows better hydrophilia than the CB/CF with a 70° water contact angle. Furthermore, a waterdrop dripped onto the TiO_2_@CB/CF quickly disappeared in 2.3 s, which is faster than the CB/CF (4.6 s). The excellent hydrophilicity and fast wettability ensure the outstanding performance of solar-vapor generation.

Except for the fast transport of water, the excellent light absorption of the TiO_2_@CB/CF is also vital for achieving high-efficiency photothermal conversion performance. From Figure 3a, we can observe that the light absorption rate of the TiO_2_/CF, TiO_2_@CB/CF, and CB/CF with the broadband wavelength range (250–2500 nm) are 51%, 92%, and 94%, respectively. The light absorption rate of the TiO_2_@CB/CF is slightly lower than the CB/CF. For the TiO_2_@CB/CF, the light absorption rate of 92% can still ensure an excellent photothermal conversion efficiency. Furthermore, the CB nanoparticles can ensure an increase in the light absorption rate of TiO_2_, improving the photocatalytic effect of TiO_2_@CB/CF.

With ultrahigh broad light absorption and the outstanding performance for water transport, the TiO_2_@CB/CF evaporator shows high efficiency in the process of solar-vapor generation. To evaluate the ability of the solar-vapor conversion of the TiO_2_@CB/CF evaporator, an experimental setup was constructed (Appendix A). It should be noted that the liquid of this experiment is MB solution, which is one of the most common pollutants in dyeing wastewater. Figure 3b shows that the surface temperature of the TiO_2_@CB/CF is 18.2 °C before irradiation. With illumination, the surface temperature of the TiO_2_@CB/CF evaporator can reach 47.2 °C within only 3 min and then maintains a stable state. The cotton fabric, TiO_2_/CF, and CB/CF also reached a steady-state temperature after 3 min. The dynamic surface temperature of the pristine cotton fabric, TiO_2_/CF, CB/CF, and TiO_2_@CB/CF were recorded by a thermocouple with irradiation time under one-sun irradiation. As demonstrated in Figure 3c, the surface temperature with steady-state for the pristine cotton fabric, TiO_2_/CF, CB/CF, and TiO_2_@CB/CF under one-sun illumination was 33.1 °C, 36.2 °C, and 48.7 °C, respectively. The temperature gap of the pristine cotton fabric, TiO_2_/CF, CB/CF, and TiO_2_@CB/CF were 15.3 °C, 16.1 °C, 30.4 °C, and 29 °C, respectively.

To explore the solar-vapor generation performance of the prepared evaporator, we measured the cumulative weight loss of water with illumination time under one-sun illumination. Figure 3d displays the liquid mass change with illumination time for the cotton fabric, TiO_2_/CF, CB/CF, and TiO_2_@CB/CF. All samples are in stable solar evaporation because the mass change of the MB solution is a linear relationship with the illumination time. The real-time water evaporation rates of all evaporators were calculated and are presented in Figure 3e. With irradiation for 30 min, the stable evaporation rate of the pristine cotton fabric, TiO_2_/CF, CB/CF, and TiO_2_@CB/CF is 0.66, 0.96, 1.37, and 1.42 kg m^−2^ h^−1^, respectively. The outstanding evaporation rate of the TiO_2_@CB/CF under illumination is due to the better thermal localization and fast water transport. It should be noted that, although the ability of light absorption for CB/CF is better than for TiO_2_@CB/CF, the evaporation rate is much lower, which indicates that the excellent performance of solar-vapor generation is not only affected by the light absorption rate but also by other factors, such as the hydrophilicity of the materials.

The efficiency of solar-vapor conversion of the sample was calculated by the following Equation (1):(1)$$\eta =mh_{LV}/P_{in}$$
where η is the efficiency of solar-vapor conversion, m is the mass flux of vapor, $h_{LV}$ denotes the total enthalpy of the water in the samples, including the phase change enthalpy (2171.2 J g^−1^ at 47.2°C for the TiO_2_@CB/CF, the detailed calculation shown in Appendix A) and the sensible heat, and $P_{in}$ is the sunlight illumination energy.

The calculation shows that the conversion efficiency of TiO_2_@CB/CF can reach 90.4% (Figure 3e), which is much higher than that of cotton fabric (44.9%), TiO_2_/CF (63.4%), and CB/CF (87.5%). This conversion efficiency of the TiO_2_@CB/CF is among the highest in current research regarding bifunctional materials when simulated dyeing water is used as an experimental solution [25,26].

With the simulated solar illumination, the photocatalytic performance of different samples was evaluated through photodegradation experiments of the MB solution. Figure 4a shows a curve of the concentration of MB solution of different samples with irradiation time. We found that the concentration curves of different samples show two types. Cotton fabric and CB/CF show a horizontal line with a tiny upward trend. The reason for this is that cotton fabric and CB/CF have no photocatalytic properties, and the concentration of MB had an increasing trend with the evaporation of water. On the contrary, because of the excellent photocatalytic effect of TiO_2_, the concentration curve of TiO_2_/CF and TiO_2_@CB/CF show a clear trend of declining with irradiation time, and the photocatalytic degradation ability of the TiO_2_/CF (~48%) is worse than that of the TiO_2_@CB/CF (~57%) after 120 min illumination. The reason for the better photocatalytic effect of TiO_2_@CB/CF includes two parts: (1) CB can change the narrow band of TiO_2_, thus expanding the catalytic absorption band; (2) CB can effectively prevent the recombination of the hole-electron pairs of TiO_2_, thus significantly increasing the life span of the hole-electron pairs [35]. In addition, the UV–vis absorption spectra of the MB solution with TiO_2_@CB/CF under different illumination times are recorded in Figure 4b. The strongest absorption peak appeared at 664 nm, which resulted from MB absorption, and under the photocatalytic effect of the TiO_2_@CB/CF, this peak has a gradually weakened trend with irradiation time. The UV–vis absorption spectra of the MB solution with cotton fabric and other functional fabric can be found in Appendix A. The peaks of the cotton fabric and CB/CF gradually increased, and the peak of the TiO_2_/CF also showed a weakening trend. These differences also verify the trend of different lines, as seen in Figure 4a.

The TiO_2_@CB/CF also shows outstanding reusability. Figure 4c demonstrates that the photocatalytic degradation performance maintains a stable level, even when the sample is reused for eight cycles. The durability of the TiO_2_@CB/CF also exhibited an excellent prospect for practical application. More importantly, from Appendix A, we can see that the photothermal layer of TiO_2_@CB is separated from the bulk water by a thermal insulating layer (PS foam). It is not directly in contact with the dyeing water. Therefore, the loss of nanoparticles in the bulk water can be prevented by this type of configuration. Moreover, the bifunctional design of the TiO_2_@CB/CF can harvest different qualities of water purification. With sunlight illumination, on the one hand, the steam generated by solar-driven evaporation can be condensed and recovered to obtain pure water. The absorbance study indicated that the condensed pure water contains no MB. On the other hand, the partially purified water remaining in the beaker still has ~43% MB in the solution. However, the concentration and color are much lower than the contaminated water (Figure 4d). Therefore, different qualities of water can be obtained by changing the irradiation time to meet different application requirements.

Actual industrial dyeing wastewater was used to explore the water purification capacity of the TiO_2_@CB/CF. The CB/CF was prepared as the controlled sample. The evaporation rates of the TiO_2_@CB/CF and CB/CF were tested under one-sun illumination in dyeing wastewater (Figure 5a). The water evaporation rates of the two samples were comparable in the first 2 h, showing excellent solar-vapor generation performance. However, in the next evaporation time, the evaporation rate of the CB/CF has a declining trend, while the TiO_2_@CB/CF still obtains a stable evaporation performance. The evaporation rate of the TiO_2_@CB/CF is much higher than that of the CB/CF. The main reason for this phenomenon is the existence of catalysis in the TiO_2_@CB/CF. For the CB/CF, in the process of solar-vapor generation, the concentration of dyeing wastewater has an upward trend, so it is more difficult for dyeing wastewater to produce vapor. On the contrary, for the TiO_2_@CB/CF, due to the catalytic effect of TiO_2_, dyeing wastewater can maintain a relatively changeless concentration, and it still maintains a stable evaporation rate. To further investigate the role of TiO_2_ and the effect of MB concentration variation, we prepared an MB solution with different concentrations (5, 10, and 20 mg L^−1^) as the treatment fluid to explore the solar-vapor generation performance of TiO_2_@CB/CF and CB/CF (Appendix A). Figure 5b shows an optical images of different dyeing wastewaters: raw wastewater, treated with CB/CF for 8 h (T-CB/CF), treated with TiO_2_@CB/CF for 8 h (T-TiO_2_@CB/CF), and purified water, respectively. We can see that the color of the T-CB/CF is deeper than the dyeing wastewater, indicating that the concentration of T-CB/CF becomes higher due to the evaporation of the water. Additionally, due to the photocatalytic effect of TiO_2_@CB/CF during solar-vapor generation, the color of T-TiO_2_@CB/CF is lighter than dyeing wastewater.

To further verify the bifunctional effect of the T-TiO_2_@CB/CF for actual industrial dyeing wastewater, the salinity of some ions and the COD of dyeing wastewater, T-CB/CF, T-TiO_2_@CB/CF, and purified water were tested. From Figure 5c, we can see that the salinity of all ions for the T-CB/CF is higher than for dyeing wastewater and the T-TiO_2_@CB/CF is similar to dyeing wastewater; the same result can be found for COD in Figure 5d. Again, these results can confirm the monofunctional water evaporation of CB/CF and the bifunctional role of TiO_2_@CB/CF. Importantly, the color, salinity, and COD of purified water meet the standard of drinking water, which could solve the problem of the current shortage of freshwater. It should be noted that, compared with dyeing wastewater, the concentrations of Cr^6+^ and Cr^3+^ in T–TiO_2_@CB/CF show a downward and upward trend, respectively. The possible reason for this phenomenon is that, under the catalysis of the TiO_2_@CB/CF, Cr^6+^ was transformed into Cr^3+^ with a reduction reaction [28,49,50,51,52]. The photocatalytic mechanism of TiO_2_@CB/CF is suggested in Figure 5e. The contaminants are absorbed into the cotton fabric. Then, HO^·^ radicals generated from holes and surface-adsorbed H_2_O (Equation (2)) would oxidize adsorbed contaminants into their oxidative products (Equation (3)). The photogenerated electrons from CB decolorize contaminants into reductive products via a chemical reduction (Equation (4)). The catalytic efficiency relates to the prevention of recombining between electrons (e^−^) and holes (h^+^) of electron−hole pairs. As a result, contaminants are reduced by the free electrons to form reductive products [53,54].
h^+^ + H_2_O → HO + H^+^(2)
Contaminants + HO^·^ → Oxidation products(3)
Contaminants + e^+^ → Reductive products(4)

In addition, the reusability of the TiO_2_@CB/CF was investigated in dyeing wastewater under one-sun illumination (Figure 5f). It retains a high evaporation rate and high efficiency, even after 20 cycles, and demonstrates great potential in practical applications.

## 4. Conclusions

In summary, a solar-driven bifunctional cotton fabric with solar-vapor generation performance and photocatalytic effect was designed and fabricated for purifying dyeing wastewater. In this novel system, the mechanism of the TiO_2_@CB/CF has two functions: (1) Owing to the photothermal ability of CB and the water transport of hydrophilic TiO_2_@CB/CF, an outstanding solar-vapor generation efficiency is achieved by thermal localization at the air–water interface; (2) TiO_2_ absorbs the UV portion of sunlight and reacts with contaminants to produce small non-toxic molecules in the process of photocatalytic decontamination; it is worth noting that the combination of CB with TiO_2_ can promote the separation of photogenerated electron-hole pairs and decrease charge recombination for the high photocatalytic activity of TiO_2_ under visible light. This bifunctional TiO_2_@CB/CF evaporator can achieve a high interfacial solar evaporation rate (1.42 kg m^−2^ h^−1^) and high solar-vapor conversion efficiency (90.4%) in MB solution under one-sun irradiation. Benefiting from the synergy effect of the photothermal effect of CB and photocatalysis of TiO_2_, the bifunctional cotton fabric demonstrates a high photocatalytic activity, with 57% of MB solution degraded after 2 h of light irradiation, and it still shows good photocatalysis, even if when reused in an MB solution for eight cycles. Moreover, the TiO_2_@CB/CF delivers excellent performance for actual industrial dyeing wastewater. Compared to the evaporator with a single functionality, TiO_2_@CB/CF provides a new approach to wastewater treatment and water purification. The novel method also offers an alternative strategy to enhance solar energy conversion and utilization.

## Figures and Tables

**Figure 1 materials-15-05917-f001:**
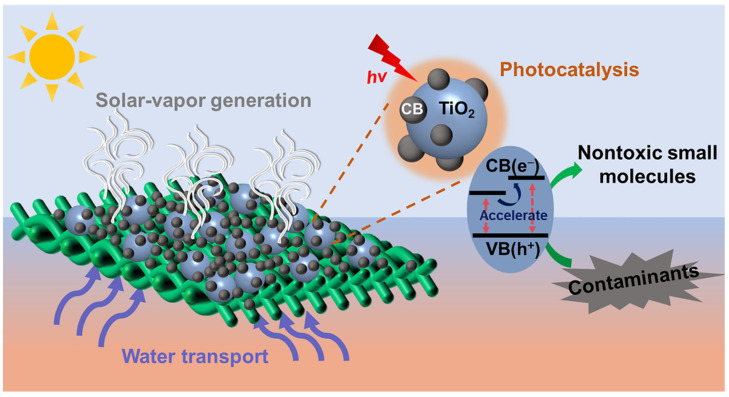
Schematic diagram of the TiO_2_@CB/CF evaporator for solar−vapor generation and photocatalytic effect.

**Figure 2 materials-15-05917-f002:**
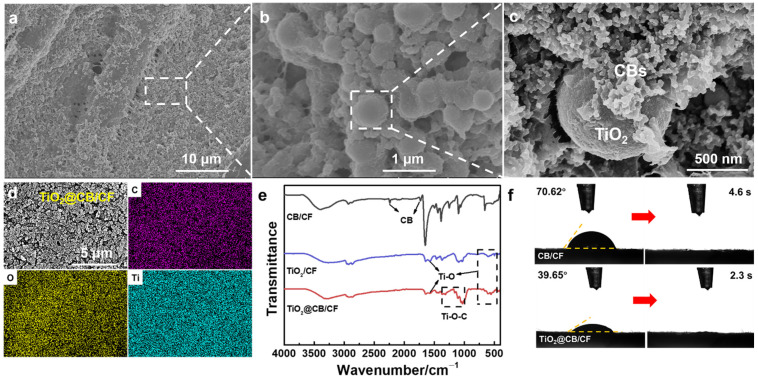
Morphologies and characterization of TiO_2_@CB/CF. (**a**–**c**) SEM images of TiO_2_@CB/CF. (**d**) EDS mapping images of TiO_2_@CB/CF. (**e**) FTIR spectra of CB/CF, TiO_2_/CF, and TiO_2_@CB/CF. (**f**) The water contact angle of CB/CF and TiO_2_@CB/CF.

**Figure 3 materials-15-05917-f003:**
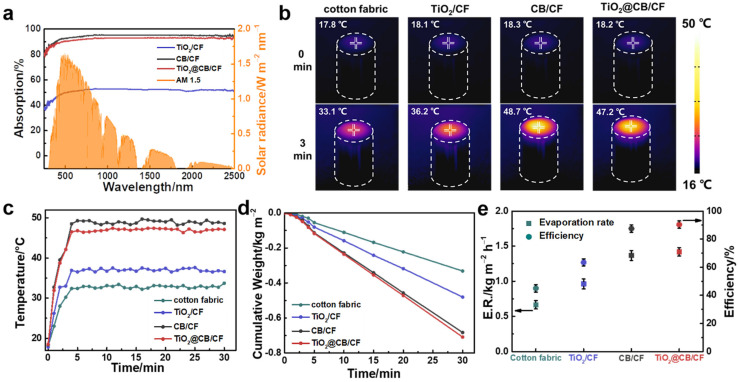
The photothermal conversion performance of the TiO_2_@CB/CF evaporator and control sample. (**a**) The light absorption curves of TiO_2_/CF, CB/CF, and TiO_2_@CB/CF. (**b**) The surface temperature of the TiO_2_@CB/CF evaporator and control sample before and after one-sun illumination by IR thermal images; (**c**) temperature evolution, (**d**) cumulative weight change, (**e**) evaporation rate evolution and energy conversion efficiency of the TiO_2_@CB/CF evaporator and control sample under 1 kW m^-2^.

**Figure 4 materials-15-05917-f004:**
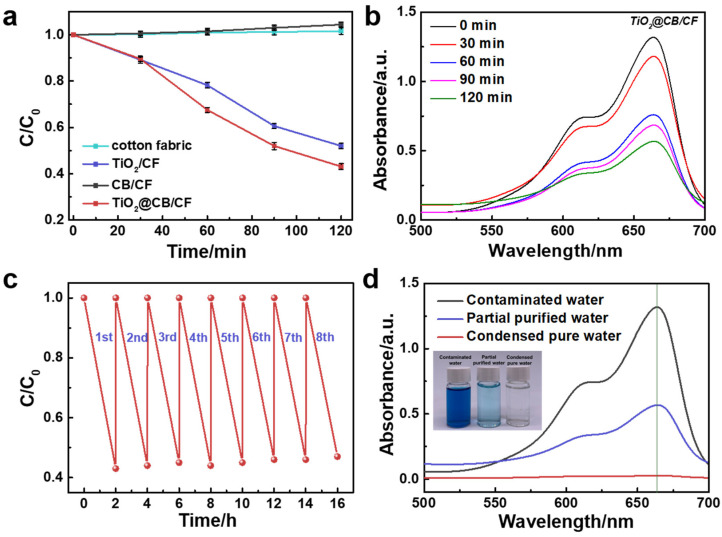
The photocatalytic ability of TiO_2_@CB/CF. (**a**) Photodegradation of MB solution with different samples for 120 min. (**b**) The UV-vis absorption spectra of the MB solution with TiO_2_@CB/CF. (**c**) The photocatalytic degradation rate of the MB solution for eight cycles. (**d**) The UV-vis absorption spectra and optical images of the MB before and after purification for 120 min by TiO_2_@CB/CF.

**Figure 5 materials-15-05917-f005:**
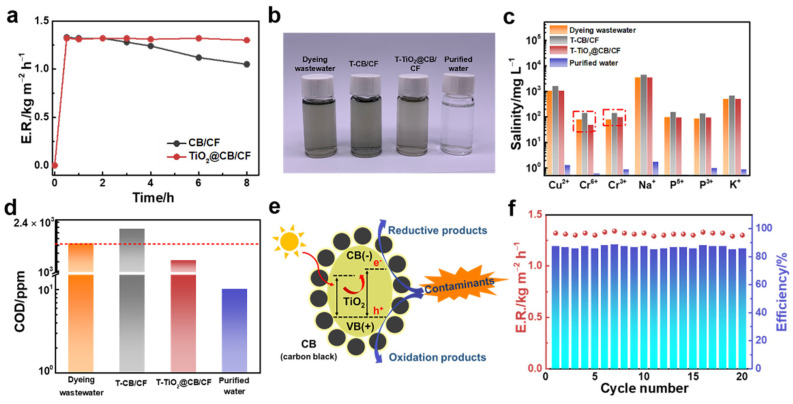
The treatment effect of actual industrial dyeing wastewater with the TiO_2_@CB/CF under one-sun illumination. (**a**) The evaporation rate of the CB/CF and TiO_2_@CB/CF with 8 h. (**b**) Optical images of dyeing wastewater, treatment with CB/CF, treatment with TiO_2_@CB/CF, and purified water. (**c**) The salinity of some ions of the industrial dyeing wastewater treatment with CB/CF, treatment with TiO_2_@CB/CF, and purified water. (**d**) The COD of the industrial dyeing wastewater treatment with CB/CF, treatment with TiO_2_@CB/CF, and purified water. (**e**) The possible photocatalytic effect of TiO_2_@CB/CF for dye degradation. (**f**) A test of the durability of the TiO_2_@CB/CF in the industrial dyeing wastewater.

## Data Availability

Not applicable.

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
