# Peer review of "A Composite Fabric with Dual Functions for High-Performance Water Purification"

_materials, 2022, doi:10.3390/ma15175917_

Round 1

Reviewer 1 Report

The paper is well-written and contributes to the field. I recommend its publication after addressing the comments below:

The contents of CB which was used in the synthesis process should be mentioned in the experimental part. What is the impact of CB concentration on the obtained results?

In page 6, the term of “worse light absorption of TiO2” should be explained based on the behaviour of TiO2 towards each wavelength band. 

What is the photothermal mechanism of CB?

Fig 4: the Y axis should be “absorbance”.

The explanation of Figure 5a is interesting, but it should be completed, and relevant references be provided. There should be more details on the role of TiO2 as the photocatalyst and the effect of MB concentration variation on the obtained results. 

It was reported that the nanoparticles are stable on cotton even after 8 cycles. However, in the absence of any binder, it seems that the nanoparticles have physically attached to the fibres. Do the authors have any estimation about the release rate of nanoparticles during the water treatment process? It can be a harmful process for environment as it is a secondary source of pollution for nature.

What is the dye adsorption capacity of TiO2@CB/CF before starting the irradiation process?

Conclusion: “TiO2 absorbs sunlight” should change to “TiO2 absorbs UV part of sunlight”.

Did CB addition show any effect on shifting the photocatalytic threshold of TiO2 nanoparticles towards visible region? 

Following papers should be cited:

https://doi.org/10.1016/j.jece.2021.106915

https://doi.org/10.1016/j.cej.2021.133688

https://doi.org/10.1016/j.colsurfa.2020.125684

Reviewer 2 Report

The manuscript from Yankuan Tian and co-workers reported the bifunctional composites deposited on cotton fabric for water purification. The paper is well written, fluent and concise in its analysis. However, it seems likely that in the figure 4(d) the solution names in the optical image are swapped. This should be verified. This article can be published after a minor revision.
